# Motivation Regarding Physical Exercise among Health Science University Students

**DOI:** 10.3390/ijerph19116524

**Published:** 2022-05-27

**Authors:** Susana Sánchez-Herrera, Javier Cubero, Sebastián Feu, Miguel Ángel Durán-Vinagre

**Affiliations:** Faculty of Education and Psychology, University of Extremadura, Avda. De Elvas S/N, 06006 Badajoz, Spain; ssanchez@unex.es (S.S.-H.); jcubero@unex.es (J.C.); sfeu@unex.es (S.F.)

**Keywords:** physical activity, self-determination theory, motivation, health science students, healthy lifestyles, well-being

## Abstract

Physical exercise and physical activity are inherent and essential agents in the evolution of active life and are associated with promoting health and well-being. This study aimed to examine the types of regulation of motivations and intentionality needed to be physically active in the future in university students from the Health Sciences branch of knowledge. Method: 351 university students with six university degrees in Health Sciences participated, 21.4% of whom were male and 78.6% women (M = 19.32; SD = 4.01). They answered the following questionnaires: “International Physical Activity Questionnaire (IPAQ)”, “Behavioural Regulation of Exercise Questionnaire (BREQ-3)”, and “Intention to be physically active (MIFAU)”. Results: intrinsic motivation and integrated regulation were positively associated (*rho* = 0.759; *p* < 0.01), as were integrated and identified codes (*rho* = 0.645; *p* < 0.01). When relating the types of motivation regulation and the intention to be physically active, men show a significant difference compared to women. It stands out that physically active people who walk are the most unmotivated (*p* < 0.01). Conclusions: more self-determined regulations and intentionality to be physically active are related to different physical activity levels and the number of METs.

## 1. Introduction

Modern-day society is changing its philosophy and how it values life, attaching greater importance to the concept of a healthy lifestyle. In this sense, the concept of active life is assumed every day by many people as an indispensable means of promoting health and well-being [1]. Physical exercise and physical activity (PA) are inherent and essential agents of this evolution, laying the foundations for any healthy lifestyle [2,3]. This transformation is backed by the importance given by the scientific community to the benefits of regular physical activity [4]. Some of them are promoted for preventing non-communicable diseases but also because of the resulting improvements in health and increased life expectancy [5]. However, despite the scientific evidence on this topic, physical inactivity, sedentary lifestyles, and high obesity rates remain among the global problems in modern society [6,7,8].

This problem is clearly reflected in the stage of adolescence, in which young adults transition from childhood to adulthood, generating a back-and-forth of changes at the biological, psychological, and social levels [9]. Likewise, this period constitutes a significant stage in the configuration of healthy lifestyles, so the creation of habits at these ages will depend on the active evolution in later years [10,11].

Furthermore, studies have found low percentages of PA among university students [12,13,14]. To address this problem, which affects adolescents, in particular in the university stage, PA is seen as a critical indicator for establishing and adopting healthy lifestyle practices and mitigating the possible dropouts generated due to the lack of PA [15]. The main objective is to lower the incidence of sedentary lifestyles and physical inactivity in this category [16], adopting the promotion of PA as a priority line [17], especially for subjects who prioritise other daily activities not linked to healthy lifestyle habits and which, in most cases, lead them to lose interest and abandon their activity [18,19].

Some studies have focused on the motivational processes associated with the practice of PA [17,20,21,22] since motivation is linked to the psychological cause of any action and is also positively related to the regulation of individual conduct [23,24,25]. In light of this association, it has been shown that motivation can be a determining factor in achieving a goal or purpose [26]. This means its absence can contribute to a lack of PA achievement, particularly among young people. This can lead to abandonment or failure to keep up the regular practice of PA, thereby affecting their quality of life and health [27,28,29].

One of the motivational theories that facilitates an understanding of how human motivation works in social contexts is the Self-Determination Theory (SDT) [30]. This theory establishes different types of motivation along a continuum depending on the level of self-determination, focusing mainly on the psychological level [31,32,33]: amotivation, extrinsic motivation (external regulation, introjected regulation, identified regulation, and integrated regulation), and intrinsic motivation [34,35,36]. Intrinsic motivation is the most self-determined motivation related to the need to explore one’s environment, curiosity, and the pleasure generated by doing an activity [36,37]. At the other end of the continuum is amotivation, understood as the lack of intention to act because the individual considers that they are incapable of achieving the expected results [37,38]. Individuals not motivated to do PA experience negative feelings such as incompetence, apathy, and even depression. That is so because they do not pursue social, affective, or material goals [39,40]. Organismic Integration Theory [30] is a sub-theory of SDT. The theory promotes internalisation and integration in the regulation, focusing on the different forms of extrinsic motivation and their systematisation [41]. Extrinsic motivation is divided into four categories: integrated, identified, introjected, and external regulation. The first of these, integrated regulation, is the most autonomous form of extrinsic motivation and is associated with identifying regulations that are evaluated according to the individual’s values and needs [36]. Identified regulation consists of conduct highly valued by the individual and judged to be significant [31,33]. Introjected regulation focuses on self-imposed expectations. It seeks to avoid anxiety and improve one’s ego, sense of worth, or pride [33,42]. Finally, external regulation is closely related to extrinsic motivation and consists of conduct to satisfy an external demand or due to the existence of rewards or prizes [33]. 

Different studies have investigated the differences in motivational regulations regarding the practice of PA according to sex [20,22,39,43,44]. The scientific literature has shown that in terms of PA among adolescents, there is a more intrinsic attribution of motivation among males and a more extrinsic attribution of cause among women [43,44,45]. Accordingly, some studies agree that women engage in less PA than men, indicating that their perception of their state of health is lower [46,47,48]. On the other hand, other studies also report higher amotivation among men [44,49,50]. In any case, further in-depth study of the motivation and interest of adolescents’ regarding the practice of PA is both necessary and essential to adapt the characteristics of the designs developed to meet the specific demands and interests of each population group. Therefore, promoting these activities will help reduce and alleviate levels of physical inactivity and contribute to the development of healthier lifestyles. 

This research aims to examine the types of regulation of motivations and intention to be physically active in the future among first-year students studying different university courses in the field of Health Sciences. Focusing on this branch of knowledge allows us to know these findings after the levels of restrictions and difficulties presented by the population under study, being able to see the influence of the pandemic on the interest and performance of physical activities.

The hypotheses to be tested in this study are the following: (1)Men will have more self-determined behaviours than women, with greater amotivation in women.(2)Men will have a greater predisposition to be physically active in the future than women.(3)People who are more self-determined will perform more vigorous and moderate physical activity (intrinsic, integrated and identified); amotivation is higher in people who perform walking physical activity than in those who perform moderate or vigorous activity.

## 2. Materials and Methods

### 2.1. Participants

The sample consisted of a total of 351 first-year university students studying six university degrees in the field of Health Science. Overall, 21.4% were men (n = 75) and 78.6% were women (n = 276), with an average age of 19.32 ± 4.01 years. Table 1 shows the breakdown of the characteristics of the students surveyed, according to gender and field of study.

### 2.2. Procedure

Consent was requested from both the different university departments that make up the area of Health Sciences and from all the participants who completed the questionnaire. They were informed that their participation was voluntary and anonymous by Spanish Law 15/1999 of 13 December on Data Protection. The ethical guidelines and codes treated all the participants according to the American Psychological Association guidelines [51]. Before handing out the questionnaires, the purpose of the study was explained, and the participants were told they would need approximately 15–20 min to complete the questionnaire. At least one researcher was present in the classroom to collect the questionnaires, and none of the participants reported difficulties completing the instrument.

### 2.3. Characteristics of the Questionnaires

A questionnaire with socio-demographic questions was administered to determine the common characteristics of the study population. In addition, the International Physical Activity Questionnaire (IPAQ) designed by [52] was administered, which consists of seven questions regarding the frequency, duration, and intensity of PA (moderate and intense) in the preceding seven days, along with walking and sitting time during a working day. The questionnaire may be administered by a self-administered survey, and it is designed for use with adults aged 18–65 years. The short version, consisting of seven items with information on the time of the individual’s time of moderate and vigorous-intensity activities, walking, and sitting, is especially recommended when the research includes population monitoring. The weekly activity is recorded in Mets (Metabolic Equivalent of Task or Metabolic Index Units) per minute per week [49]: walking: 3.3 Mets; moderate physical activity: 4 Mets; vigorous physical activity: 8 Mets.

A validated Spanish version of the Behavioural Regulation in Exercise Questionnaire—BREQ-3 [35,50] was also included. The BREQ-3 is made up of 23 items grouped into six factors that begin with the phrase “I do exercise…” The motivation factors are: intrinsic (four items, e.g., “Because I think exercise is fun”, with an α = 0.89), integrated (four items, e.g., “Because it suits my lifestyle”, with an α = 0.90), identified (three items, e.g., “Because I value the benefits of physical exercise”, with an α = 0.78), introjected (four items, e.g., “Because I feel guilty when I don’t exercise”, with an α = 0.69), external (four items, e.g., “Because others tell me I should do it”, with an α = 0.71), and amotivated (four items, e.g., “Because I don’t see why I have to do it”, with an α = 0.80). The values of Cronbach’s alpha were mostly adequate (α > 0.70) [53].

Finally, the instrument also included the Questionnaire for the Measurement of Intention to be Physically Active in the University Context (MIFAU), based on the Spanish version by [54]. This questionnaire starts with the phrase, “Regarding your intention to engage in physical-sports activity…” and consists of five items. The answers consist of a Likert-type scale ranging from 1 to 5, where one corresponds to “strongly disagree” and five to “strongly agree”. The reliability of the instrument yielded α = 0.80 and was therefore adequate (α > 0.70) [53].

### 2.4. Statistical Analysis

To determine the nature of the data, the statistical programme SPSS 25 (Statistical Package for the Social Sciences, IBM Corp. Released 2012. IBM SPSS Statistics for Windows, Version 25, IBM Corp., Armonk, NY, USA) was used. Subsequently, a descriptive and correlational analysis was carried out to determine the relationships between the variables studied. 

The responses obtained from each of the questionnaires (IPAQ, BREQ-3, and MIFAU) were analysed to assess the adequacy of the model [55]: the global goodness of fit index (GFI), the incremental fit index (IFI), the comparative fit index (CFI), the normed fit index (NFI), the standardised root mean square residual (SRMR), and the root mean square error of approximation (RMSEA). The CFI and GFI values range from 0 to 1, with 0 indicating no fit and 1 indicating an optimal fit. Values of 0.95 or above are considered excellent, and values above 0.90 suggest an acceptable fit of the model to the data. The RMSEA index is considered optimal when its values are 0.05 or lower and acceptable when the range is between 0.08–0.05 [56,57]. The reliability of the questionnaires was also calculated using Cronbach’s alpha, considering α > 0.70 [53] as appropriate factors.

## 3. Results

A confirmatory factor analysis (CFA) of the BREQ questionnaire showed adequate fit indices: χ2/gl = 2.20, GFI = 0.90, IFI = 0.94, CFI = 0.94, RMSEA = 0.056, SRMR = 0.070, PClose = 0.085 (*p* > 0.05). The MIFAU questionnaire showed excellent fit indices: χ2/gl = 1.07, GFI = 0.99, IFI = 1.00, CFI = 1.00, RMSEA = 0.014, SRMR = 0.016, PClose = 0.67 (*p* > 0.05). The reliability of the scales used was adequate (α > 0.70). The Kolmogorov-Smirnov test (K-S) includes the Lillefors correction, meaning that none of the variables analysed comply with the principle of normality. 

In the BREQ-3 variables, the means obtained for intrinsic motivation and integrated regulation are the highest, scoring at 2.79 ± 1.03 and 2.28 ± 1.16, respectively (Table 2). In contrast, external motivation and amotivation have the lowest scores, with means of 0.34 ± 0.56 and 0.41 ± 0.70, respectively. Similarly, university students gave the MIFAU a mean value of 3.88 ± 0.83, with the maximum score of the questionnaire being five points. In the IPAQ, the values were 2277.82 ± 2258.03 for the metabolic index units.

The integrated regulation scores show a high and significant correlation with intrinsic motivation (*rho* = 0.759; *p* < 0.01). The correlation between the identified regulation and integrated regulation is noteworthy (*rho* = 0.645; *p* < 0.01). In contrast, amotivation correlates negatively with intrinsic motivation (*rho* = −0.450; *p* < 0.01) and integrated regulation (*rho* = −0.418; *p* < 0.01). 

The participants correlated positively with intention to be physically active and intrinsic (*rho* = 0.678; *p* < 0.01), integrated (*rho* = 0.734; *p* < 0.01), identified (*rho* = 0.532; *p* < 0.01), and introjected (*rho* = 0.233; *p* < 0.01) motivation, being associated with the most positive levels of self-determination. There was also a high and significant correlation between amotivation and intention to be physically active (*rho* = −0.454; *p* < 0.01).

Finally, the IPAQ correlates positively with the mean values for the MIFAU (*rho* = 0.414; *p* < 0.01), integrated regulation (*rho* = 0.391; *p* < 0.01) and intrinsic motivation (*rho* = 0.333; *p* < 0.01). However, it correlates negatively with amotivation (*rho* = −0.232; *p* < 0.01).

Motivational regulation and intention to be physically active were analysed according to the participant’s sex. Results showed that men had a statistically significant difference (*p* < 0.01) greater than women in terms of intrinsic and integrated regulation of PA and greater intention to be physically active in the future (Table 3). 

Table 4 shows the differences in the regulation of motivations and intention to be physically active according to the academic degree. No differences in external regulation are observed (*p* > 0.05).

Based on the results of the IPAQ questionnaire regarding walking, moderate, and vigorous PA [52]. Subsequently, motivation was analysed for the PA classification for university students. The inferential results indicate statistically significant differences for the following variables: intrinsic, integrated, identified, amotivation, and intention to be physically active (*p* < 0.01) (Table 5). 

The multiple pairwise comparisons (Figure 1) show that those who engage in vigorous activity have a higher integrated intrinsic regulation, as well as a higher intention to be physically active than those who engage in walking or moderate activity (*p <* 0.01). Those who engage in vigorous or moderate PA have a higher identified regulation than those who engage in walking activities (*p <* 0.01). People who engage in walking activities are the most amotivated (*p <* 0.01). 

## 4. Discussion

This study sought to examine the types of regulation of motivation and intention to be physically active in the future among first-year students studying different university degrees in the field of Health Sciences. Our results indicate that health science students have a more favourable tendency toward intrinsic motivation and integrated regulation, with lower scores for external stimulation and inspiration. These data are in line with those reported by [39] as they show the same differences in terms of mean comparisons. The same was established by [58], which obtained similar data to ours when comparing these variables for a sample of teacher trainees.

As for the degree of relationship between the variables analysed, we found that intrinsic motivation and integrated regulation were positively associated, as were integrated and identified codes. These data are in line with the results of [59], as they found associations between the most autonomous regulations (intrinsic, integrated, and identified). However, external code and amotivation are positively and significantly related to each other and negatively associated with the rules above. These results are similar to those of other related studies [39,60]. The factors at the end of the continuum correlate positively and with higher scores, as shown by several studies [20,39,50]. 

Related to our findings, other authors have investigated physical activity with other branches of knowledge different from Health Sciences, as is the case of [61]. However, few studies have been found in relation to physical activity with future professionals related to the field of Health Sciences, highlighting [62], emphasising the importance of addressing this issue even more. It stands out in our results that students more linked to the area of Social Sciences (Psychology and Occupational Therapy) are those who practice less physical activity. Regarding the intention to be physically active in the future, the results indicate that it correlates positively and significantly with intrinsic motivation and integrated regulation. These more self-determined forms predictably indicate the intention to be physically active [63,64,65]. Integrated regulation seemed to be more critical among university students than the intrinsic motivation to become more physically active in the future [35]. Similarly, the correlational study shows that the intention to be physically active has a negative relationship with amotivation, similar to the findings of other studies [66,67].

On the other hand, when relating the amount of METs to the types of motivation, the results show that the amount of METs correlates positively and significantly with intrinsic motivation, integrated regulation, and identified code, but negatively with amotivation. Similar results were found in the study by [68], which highlighted the direct and inverse correlations for the totality of the METs and the correlation analysis with the most self-determined form and amotivation. Likewise, other authors determined the importance of developing motivational processes to improve physical activity levels, especially in less related degrees [17].

Regarding the sex differences, our research shows that there are changes in the motivation and achievement of PA among university students, with men having higher values than women in all types of regulation (intrinsic, integrated, identified, introjected, and external). Similarly, in the specific intrinsic and integrated regulation cases, men had significantly higher values than women. These results are similar to those reported by [69], who indicate that women engage in less PA than men. Another study that confirms our research results for Health Science students is the recent work published by [70], in which the main differences regarding motivation between women and men were that for men, the most important reason for engaging in PA was the pleasure of doing it, while women mainly engaged in PA due to the desire to maintain a good state of health. These results are manifested and reflected in a reduction in the PA indices that differs according to sex, with a lower impact for women [28,71].

Finally, when relating the number of METs to the types of motivation, the results show that the number of METs correlates positively and significantly with intrinsic motivation, integrated regulation, and identified code, and negatively but significantly with basis. These results were similar to the study by [68], which confirmed a direct and inverse correlation for the totality of the METs and correlation analysis with the most self-determined form and amotivation. Our results highlight the importance of the most self-determined regulation types (intrinsic, integrated, and identified), inspiration, and intention to be physically active for the different levels of students’ PA (walking, moderate, and vigorous), with our data showing a significant relationship. These results are consistent with those found in other studies of university students [64,68,72,73].

It should be noted that, with the development of this work, as a limitation and a potential for a future line of research, it is necessary to indicate the social and economic status of university students. Likewise, the possibility of using accelerometers to specify the type of physical activity further could be considered. 

As practical implications, physical activities within the university campus itself can be used and clarified as strategies, establishing flexible schedules for students to attend the activities offered by the university, linking physical activity information with health maintenance, and analysing the causes of dropout during the university stage.

## 5. Conclusions

The results obtained in the study confirm that the factors with higher self-determination correlate positively and significantly with each other and inversely with those at the end of the continuum. Similarly, university students studying different Health Science undergraduate degrees show changes in motivation and PA according to sex, with statistically significant differences when comparing both groups in the case of intrinsic motivation and integrated regulation and higher values for men than for women. 

The importance granted to more self-determined regulation types and the intention to be physically active in the future is related to the different levels of PA and the number of METs. This reflects the critical period these university students are going through, characterised by a general decrease in PA levels, which places them at a disadvantage in maintaining those levels. Therefore, based on our findings, PA intervention programmes for university students should be focused and targeted on both the interests and preferences of this population group at the individual or collective level. Activities should also be carried out on the university campus to make them compatible with studies, making the timetables more flexible and adaptable. This initiative will involve raising awareness of the possible options to engage in PA on the university campus and the resulting health benefits. We must show the way to transform habits and customs so that individuals appreciate the valuable benefits of PA, which ultimately lead to a better quality of life. 

## Figures and Tables

**Figure 1 ijerph-19-06524-f001:**
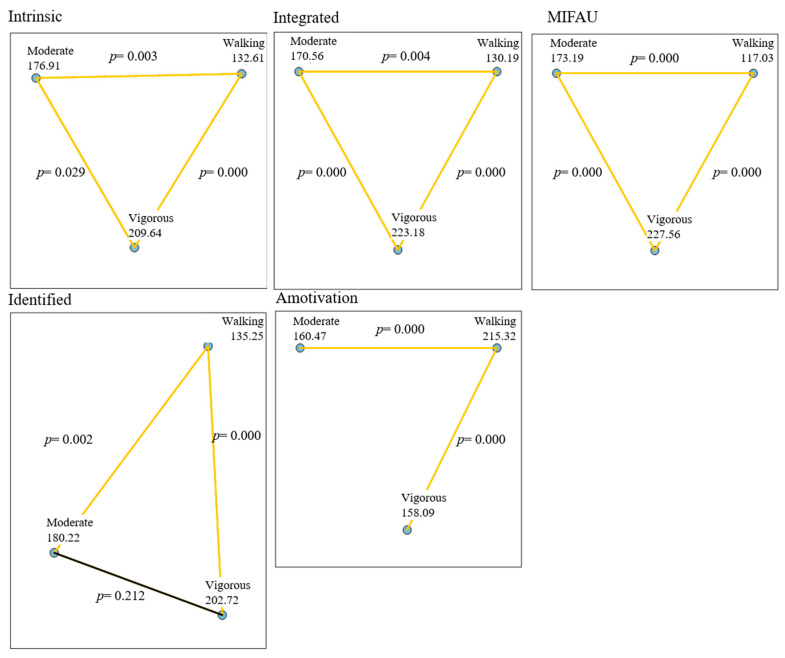
Multiple pairwise comparisons of the behavioural regulation in exercise variables according to the Physical Activity.

**Table 1 ijerph-19-06524-t001:** Characteristics of the sample according to the field of study.

Bachelor’s Degree	Male	Women
Physiotherapy	30.2% (n = 16)	69.8% (n = 37)
Nursing	18.8% (n = 12)	81.3% (n = 52)
Medicine	27.7% (n = 26)	72.3% (n = 68)
Psychology	9.7% (n = 6)	90.3% (n = 56)
Occupational Therapy	17.4% (n = 4)	82.6% (n = 19)
Veterinary Medicine	20% (n = 11)	80% (n = 44)

**Table 2 ijerph-19-06524-t002:** Descriptives of the study.

	α	M	SD	Variance	Skewness (Desv. Err = 0.130)	Kurtosis (Desv. Err = 0.260)	K-S	I	II	III	IV	V	VI	VII
I. Intrinsic	0.89	2.79	1.03	1.06	−0.85	0.099	0.140 **							
II. Integrated	0.90	2.28	1.16	1.35	−0.16	−1.04	0.090 **	0.759 **						
III. Identified	0.78	2.17	0.68	0.47	−0.98	1.03	0.215 **	0.544 **	0.645 **					
IV. Introjected	0.69	1.11	0.88	0.78	0.69	−0.18	0.124 **	0.159 **	0.298 **	0.411 **				
V. External	0.71	0.34	0.56	0.31	1.95	3.55	0.330 **	−0.239 **	−0.112 *	0.279 **	0.342 **			
VI. Amotivation	0.80	0.41	0.70	0.48	2.10	4.68	0.320 **	−0.450 **	−0.418 **	−0.365 **	−0.021	0.237 **		
VII. MIFAU	0.80	3.88	0.83	0.70	−0.90	0.36	0.140 **	0.678 **	0.734 **	0.532 **	0.233 **	−0.157 **	−0.454 **	
IPAQ-Mets	-	2277.82	2258.03	5,098,684.95	2.46	10.34	0.157 **	0.333 **	0.391 **	0.312 **	0.145 **	−0.021	−0.232 **	0.414 **

* *p* < 0.05; ** *p* < 0.01; M: Mean; SD: Standard Deviation.

**Table 3 ijerph-19-06524-t003:** Regulation of motivation and intention to engage in PA according to sex.

		M	SD	X^2^	*p*	E^2^_R_
Intrinsic	Man	3.16	0.86	7532.00	0.000 **	0.04
	Woman	2.69	1.05			
Integrated	Man	2.79	1.15	6981.50	0.000 **	0.05
	Woman	2.15	1.13			
Identified	Man	2.27	0.60	9172.50	0.123	0.007
	Woman	2.14	0.70			
Introjected	Man	1.20	0.99	9898.00	0.560	0.001
	Woman	1.08	0.85			
External	Man	0.37	0.67	10171.00	0.794	0.000
	Woman	0.33	0.53			
Amotivation	Man	0.39	0.63	10149.00	0.771	0.000
	Woman	0.42	0.71			
MIFAU	Man	4.13	0.75	7793.50	0.001 **	0.03
	Woman	3.81	0.84			

** *p* < 0.01.

**Table 4 ijerph-19-06524-t004:** Regulation of motivation and intention to engage in PA according to academic degree.

	X^2^	*p*	E^2^_R_	Pairwise Comparisons
Intrinsic	22.213	0.000 **	0.05	Medicine > Occupational Therapy Medicine > Psychology Physiotherapy > Psychology
Integrated	35.204	0.000 **	0.088	Medicine > Occupational Therapy Medicine > Psychology Medicine > Veterinary Medicine Physiotherapy > Psychology
Identified	27.427	0.000 **	0.065	Medicine > Occupational Therapy Medicine > Psychology Physiotherapy > Occupational Therapy Physiotherapy > Psychology
Introjected	15.592	0.000 **	0.031	Nursing > Occupational Therapy Physiotherapy > Occupational Therapy
External	6.770	0.238	0.005	-
Amotivation	12.550	0.028 *	0.022	Medicine > Psychology
MIFAU	31.696	0.000 **	0.077	Medicine > Psychology Physiotherapy > Psychology Nursing > Psychology

* *p* < 0.05; ** *p* < 0.01.

**Table 5 ijerph-19-06524-t005:** Regulation of motivation and intention to engage in PA according to the PA practised.

Variable BREQ	Physical Activity	M	SD	X^2^	*p*	E^2^_R_
Intrinsic	Walking	2.32	1.13	28.08	0.000 **	0.07
	Moderate	2.82	0.98			
	Vigorous	3.12	0.87			
Integrated	Walking	1.72	1.09	44.15	0.000 **	0.12
	Moderate	2.23	1.09			
	Vigorous	2.82	1.09			
Identified	Walking	1.88	0.76	22.71	0.000 **	0.06
	Moderate	2.20	0.66			
	Vigorous	2.36	0.57			
Introjected	Walking	0.96	0.87	5.31	0.070	0.01
	Moderate	1.09	0.81			
	Vigorous	1.26	0.97			
External	Walking	0.36	0.54	2.67	0.263	0.002
	Moderate	0.36	0.58			
	Vigorous	0.285	0.54			
Amotivation	Walking	0.67	0.79	22.60	0.000 **	0.06
	Moderate	0.35	0.62			
	Vigorous	0.30	0.67			
MIFAU	Walking	3.37	0.90	57.79	0.000 **	0.16
	Moderate	3.89	0.74			
	Vigorous	4.26	0.69			

** *p* < 0.01.

## Data Availability

The data presented in this study are available on request from the corresponding author.

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
