# Peer review of "Motivation Regarding Physical Exercise among Health Science University Students"

_ijerph, 2022, doi:10.3390/ijerph19116524_

Round 1

Reviewer 1 Report

The authors provide a great manuscript that has lots of merit in the field. I do have some suggestions for the authors to improve the manuscript. 

Introduction:

In line 27 the authors state "Proof of this..." However, in science, we never 'prove' anything rather just provide evidence to justify the findings. I believe this language should be changed. 

There are a few run-on sentences in the introduction such as lines 31-34, and 40-44.

Additionally, lines 38-44 discuss adolescents and PA which is not what the study is about. This study is focused on the population of college-aged individuals, therefore, the introduction should contain information about the population at hand. 

Materials and Methods

Lines 95-96 should be at the top of the next page. 

The authors discuss the IPAQ questionnaire in lines 106-117. They mention the questionnaire could be administered by direct interview, telephone, or self. Which one was administered in this study? I think clarifying this would help the understanding of what the procedure actually was.

Suggestion to move 'Procedure' section right after 'Subjects' then followed by the 'Instruments' for more clarity. 

Results

In line 166 the authors present the 'FBA' of the BREQ questionnaire, however, (unless I'm missing it in the preceding paragraphs), there is no indication of what FBA is, which should be stated. 

Discussion

In lines 221-223 the authors indicate what university the study took place when they did not state this in the methods. I would remove the school where the study took place. Also, the authors mention 'health sciences' as the field. This could be mentioned in the methods instead of listing out all the different degrees in lines 98-101. 

In line 243-245 the authors mention 'other studies' which needs a reference/citation to which studies they are referring. 

General

The authors use the term 'gender' to indicate male/female, however, I would suggest the authors change to the term 'sex'. 

The authors did a great job with this manuscript and I hope they take my suggestions. Thank you and good luck. 

Reviewer 2 Report

Dear authors,

thank you for your very interesting article. Here are my suggestions:

What does METs. stands for in the abstracts? Metabolic Equivalent of Task?

At the eginning you mentioned Both physical exercise and physical activity (PA) are 29 inherent and essential agents of this evolution but you do not write about it.

You state that further in-depth study of the motives of adolescents regarding the practice of PA is both necessary and essential in order to adapt the characteristics of the designs.

Why you havent done it?

Rewrite the abstract into: 

Physical exercise and physical activity are inherent and essential agents in the evolution of active life and are associated with promoting health and well-being. This study aimed to examine the types of regulation of motivations and intentionality needed to be physically active in the future in university students from the Health Sciences branch of knowledge. Method: 351 university students with seven university degrees in Health Sciences participated, 21.4% of whom were male and 78.6% female (M = 19.32; SD = 4.01). They answered the following questionnaires: "International Physical Activity Questionnaire (IPAQ)", "Behavioural Regulation of Exercise Questionnaire (BREQ-3)", and "Intention to be physically active (MIFAU)". Results: intrinsic motivation and integrated regulation were positively associated (rho =.759; p .01), as were integrated and identified codes (rho =.645; p .01). When relating the types of motivation regulation and the intention to be physically active, men show a significant difference compared to women. It stands out that physically active people who walk are the most unmotivated (p .01). Conclusions: more self-determined regulations and intentionality to be physically active are related to different physical activity levels and the number of METs.

Modern-day society is changing its philosophy and how it values life, attaching greater importance to the concept of a healthy lifestyle. Proof of this is the large number of people who engage in active living practices daily as an indispensable means of promoting health and well-being. Physical exercise and physical activity (PA) are inherent and essential agents of this evolution, laying the foundations for any healthy lifestyle (1, 2). This transformation is backed by the importance given by the scientific community to the benefits of regular physical activity, not only as a means of preventing non-communicable diseases but also because of the resulting health improvements and increased life expectancy (3, 4). However, despite the scientific evidence on this topic, physical inactivity, sedentary lifestyles, and high obesity rates remain among the global problems in modern society (5-7).

Furthermore, studies have found low percentages of PA among university students (8–10). To address this problem, which affects adolescents, PA is seen as a critical indicator for establishing and adopting healthy lifestyle practices and mitigating the possible dropouts generated due to the lack of PA (11). The main objective is to lower the incidence of sedentary lifestyles and physical inactivity in this category (12), adopting the promotion of  PA as a priority line (13), especially for subjects who prioritise other daily activities not linked to healthy lifestyle habits and which, in most cases, lead them to lose interest and abandon their activity (14, 15).

Some studies have focused on the motivational processes associated with P.A. practice (13, 16-18), since motivation is linked to the psychological cause of any action and is also positively related to the regulation of individual conduct (19-21). In light of this association, it has been shown that motivation can be a determining factor in achieving a goal or purpose (22). This means its absence can contribute to a lack of P.A. achievement, particularly among young people. This can lead to abandonment or failure to keep up the regular practice of P.A., thereby affecting their quality of life and health (23–51). One of the motivational theories that facilitate understanding how human motivation works in social contexts is Self-Determination Theory (SDT) (26). This theory establishes different types of motivation along a continuum depending on the level of self-determination, focusing mainly on the psychological level (27-29): motivation, extrinsic motivation (external regulation, introjected regulation, identified regulation, and integrated regulation), and intrinsic motivation (30–32). Intrinsic motivation is the most self-determined motivation related to the need to explore one's environment, curiosity, and the pleasure generated by doing an activity (32, 33). At the other end of the continuum is motivation, understood as the lack of intention to act because the individual considers that they are incapable of achieving the expected results (33, 34). Individuals not motivated to do P.A. experience negative feelings such as incompetence, apathy, and even depression. That is so because they do not pursue social, affective, or material goals (35, 36). Organismic Integration Theory (26) is a sub-theory of SDT. The theory promotes internalisation and integration in regulation, focusing on the different forms of extrinsic motivation and their systematisation (37). Extrinsic motivation is divided into three categories: integrated, identified, introjected, and external regulation. The first of these, integrated regulation, is the most autonomous form of extrinsic motivation and is associated with identifying regulations that are evaluated according to the individual's values and needs (32). Identified regulation consists of conduct highly valued by the individual and judged to be significant (27, 29).

Introjected regulation focuses on self-imposed expectations. It seeks to avoid anxiety and improve one's ego, sense of worth, or pride (29, 38). Finally, external regulation is closely related to extrinsic motivation and consists of conduct to satisfy an external demand or due to the existence of rewards or prizes (29). Different studies have investigated the differences in motivational regulations regarding the practice of P.A. according to gender (16, 17, 35, 39, 40). The scientific literature has shown that in terms of P.A. among adolescents, there is a more intrinsic attribution of motivation among males and a more extrinsic attribution of cause among females (41-43). Accordingly, some studies agree that women engage in less P.A. than men, indicating that their perception of their state of health is lower (44–46).

On the other hand, other studies also report higher motivation among men (40, 47, 84, 48). These results are manifested and reflected in a reduction of P.A. indices that differs according to gender, with a lower impact for women (24, 49). In any case, further in-depth study of adolescents' motives regarding the practice of P.A. is both necessary and essential to adapt the characteristics of the designs developed to meet the specific demands and interests of each population group. Therefore, promoting these activities will help reduce and alleviate levels of physical inactivity and contribute to the development of healthier lifestyles. This research aims to examine the types of regulation of motivations and intentions to be physically active in the future among first-year students with different university degrees at the University of Extremadura in Health Sciences. The sample consisted of a total of 351 first-year university students from different 97 university degrees, more specifically, the Bachelor's Degree in Physiotherapy, the Bachelor's Degree in Nursing, the Bachelor's Degree in Medicine, the Bachelor's Degree in Psychology, the Bachelor's Degree in Occupational Therapy, and the Bachelor's Degree in Veterinary Medicine. 21.4% were men (n = 75), and 78.6% were women (n = 276), with an average age of 19.32±4.01 years.  Instruments A questionnaire with socio-demographic questions were administered to determine the common characteristics of the study population. In addition, the International Physical Activity Questionnaire (IPAQ) designed by (50) was administered, which consists of seven questions regarding the frequency, duration, and intensity of PA (moderate and intense) in the preceding seven days, along with walking and sitting time during the working day. The questionnaire may be administered by direct interview, telephone, or self-administered survey, and it is designed for use with adults aged 18-65 years. The short version, consisting of seven items with information on the individual's time on moderate and vigorous-intensity activities, walking, and sitting, is especially recommended when the research includes population monitoring. The weekly activity is recorded in Mets (Metabolic Equivalent of Task or Metabolic Index Units) per minute per 115 weeks: 3.3 Mets; moderate physical activity: 4 Mets; vigorous physical activity: 8 Mets. A validated Spanish version of the Behavioural Regulation in Exercise Questionnaire -  BREQ-3 (31, 51) was also included. The BREQ-3 is made up of 23 items grouped into six factors that begin with the phrase "I do exercise..." The motivation factors are: intrinsic  (four items, e.g., "Because I think exercise is fun", with an α = 0.89), integrated (four items, e.g., "Because it suits my lifestyle", with an α = 0.90), identified (three items, e.g., "Because I value the benefits of physical exercise", with an α = 0.78), introjected (four items, e.g., "Because I feel guilty when I don't exercise", with an α = 0.69), external (four items, e.g., 124 "Because others tell me I should do it", with an α = 0.71), and motivated (four items, e.g., 125 "Because I don't see why I have to do it", with an α = 0.80). The values of Cronbach's alpha 126 were mostly adequate ( > 0.70) (52). 127 Finally, the instrument also included the Questionnaire for the Measurement of Intention to be Physically Active in the University Context (MIFAU), based on the Spanish version by 129 (53). This questionnaire starts with the phrase, "Regarding your intention to engage in physical-sports activity..." and consists of five items. The answers consist of a Likert-type scale ranging from 1 to 5, where one corresponds from "strongly disagree" to "strongly agree." The reliability of the instrument yielded α = 0.80 and was therefore adequate ( >.70) (52). Procedure  Consent was requested from both the different university departments that make up the area of Health Sciences and from all the participants who completed the questionnaire. They were informed that their participation was voluntary and anonymous by Spanish Law 15/1999 of December 13 on Data Protection. The ethical guidelines and codes treated all the participants of conduct of the American Psychological Association (54). Before handing out the questionnaires, the purpose of the study was explained, and the participants were told they would need approximately 15-20 minutes to complete the questionnaire. At least one researcher was present in the classroom to collect the questionnaires, and none of the participants reported difficulties completing the instrument. Motivational regulation and the intention to be physically active were analysed according to the participant's gender.

Results showed that men had a statistically significant difference (p<0.01) greater than females in terms of intrinsic and integrated regulation of PA and a greater intention to be physically active in the future.

Based on the results of the IPAQ questionnaire, into walking, moderate, and vigorous PA (50). Subsequently, motivation was analysed for the PA classification for university students. The inferential results indicate statistically significant differences for the following variables: intrinsic, integrated, 205 identified, motivation, and intention to be physically active (p<0.01) (Table 3). The multiple pairwise comparisons (Figure 1) show that those who engage in vigorous activity have a higher integrated intrinsic regulation, as well as a higher intention to be physically active than those who engage in walking or moderate activity (p<0.01).  Those who engage in vigorous or moderate PA have a higher identified regulation than those who engage in the walking activity (p<0.01). People who engage in walking activities are the most motivated (p<0.01). This study sought to examine the types of regulation of motivation and intention to be physically active in the future among first-year students of different university degrees at the University of Extremadura in Health Sciences. Our results indicate that health science students have a more favourable tendency toward intrinsic motivation and integrated regulation, with lower scores for external stimulation and inspiration. These data are in line with those reported by (35) as they show the same differences in terms of mean comparisons. The same was established by (58), which obtained similar data to ours when comparing these variables for a sample of teacher trainees. As for the degree of relationship between the variables analysed, we found that in- intrinsic motivation and integrated regulation were positively associated, as were integrated and identified codes. These data are in line with the results of (59) and (35), as they found associations between the most autonomous regulations (intrinsic, integrated, and identified). However, external code and motivation are positively and significantly related to each other and negatively associated with the rules above. These results are similar to those of other related studies (35, 60). The factors at the end of the continuum correlate positively and with higher scores. Regarding the intention to be physically active in the future, the results indicate that it correlates positively and significantly with intrinsic motivation and integrated regulation. These more self-determined forms predictably indicate the intention to be physically active (61-63). Integrated regulation seemed to be more critical among university students than the intrinsic motivation to become more physically active in the future (31). Similarly, the correlational study shows that the intention to be physically active has a negative relationship with motivation, similar to the findings of other studies. 245 On the other hand, when relating the amount of METs to the types of reason, the 246 results show that the amount of METs correlates positively and significantly with intrinsic motivation, integrated regulation, and identified code, but negatively with motivation. Similar results were found in the study by (64), which highlighted the direct and inverse correlations for the totality of the METs and the correlation analysis with the most self-determined form and motivation. (13) determined the importance of developing motivational processes to improve physical activity levels, especially in less related degrees. Regarding gender differences, our research shows that there are changes in the motivation and achievement of PA among university students, with men having higher values than women in all types of regulation (intrinsic, integrated, identified, introjected, and external). Similarly, in the specific intrinsic and integrated regulation cases, men had significantly higher values than women. These results are similar to those reported by (65), (9), and (16), who indicate that women engage in less PA than men. Another study that confirms our research results for Health Science students is the recent work published by (39), in which the main differences regarding motivation between women and men were that for men, the most important reason for engaging in PA was the pleasure of doing it, while women mainly engaged in PA due to the desire to maintain a good state of health. Finally, when relating the number of METs to the types of motivation, the results show that the number of METs correlates positively and significantly with intrinsic motivation, integrated regulation, and identified code, and negatively but significantly with basis. The results were similar to the study by (64), which confirmed a direct and inverse correlation for the totality of the METs and correlation analysis with the most self-determined form and motivation. Our results highlight the importance of the most self-determined regulation types (intrinsic, integrated, and identified), inspiration, and intention to be physically active for the different levels of students’ PA (walking, moderate, and vigorous), with our data showing a significant relationship.

The results obtained in the study confirm that the factors with higher self-determination correlate positively and significantly with each other and inversely with those at the end of the continuum. Similarly, university students from different health science undergraduate degrees show changes in motivation and PA according to gender, with statistically significant differences when comparing both groups in the case of intrinsic motivation and integrated regulation and higher values for men than for women. The importance granted to more self-determined regulation types and the intention to be physically active in the future is related to the different levels of PA and the number of METs. This reflects the critical period these university students are going through, characterized by a general decrease in PA levels, which places them at a disadvantage in maintaining those levels. Therefore, based on our findings, PA  intervention programmes for university students should be focused and targeted on both the interests and preferences of this population group at the individual or collective level. Activities should also be carried out on the university campus to make them compatible with studies, making the timetables more flexible and adaptable. This initiative will involve raising awareness of the possible options to engage in PA on the university campus and the resulting health benefits. We must show the way to transform habits and customs so that individuals appreciate the valuable benefits of PA, which ultimately lead to a better quality of life.

Very good article.

Wish you all the luck in the future!

Reviewer 3 Report

Manuscript details:

Journal: International Journal of Environmental Research and Public Health (ISSN 1660-4601)

Manuscript ID: ijerph-1716567

Type: Article

Title: Motivation regarding physical exercise among Health Science university students

Section: Mental Health

Special Issue: Mental Health and Wellbeing in Times of Change

Comments and Suggestions for Authors

Article review

In this article, the Authors present the results of three surveys assessing the motivation to physical activity (AP) among students of the University of Extremadura in Spain. The subject of the article fits into the current social issues, emphasizes the role of health promotion and the resulting well-being through physical activity. The aim of the article was to draw attention to the important role of motivation (depending on the type) for future physical activity. The article is well organized with five main points.

Below I present my main comments.

Abstract

- What does the term - from seven University degrees (line 12) mean, are they faculties? this is what I can assume from the rest of the article - in my opinion there is one bachelor's degree obtained in various fields of study

- Does it matter to express the age with such precision, in hundredths? (lines 13 and 102), it is good to state what the abbreviation means; M and SD (why is it written - this last- once without dots and elsewhere with dots - Table 1)

- Do METs (line 21) and Mets (e.g. line 115) define the same?

Introduction

- Are the first sentence of the first paragraph and the first sentence of the second paragraph not mutually exclusive? this point of mine only points to the need for more thought on this issue, as the problem stems from the difficulty of translating scientific evidence into the daily habits of socjety

- Are there any measurement techniques to differentiate between exercise and physical activity?

- Why (line 64) there are dots after P and A?

- Would the paragraph (lines from 78 to 86) not be better moved to the discussion?

- What different university degrees does line 93-94 refer to? With this question I want to draw your attention to the unfortunate word "degree" - of course that is only my opinion

Materials and Methods

- I would suggest entering the number of surveyed students (broken down by gender) depending on the field of study

- Why do the authors mention six fields of study / faculties (lines 98 - 101) and previously was mentioned of seven?

- Could the word Instruments (line 104) be replaced with another word? maybe - Characteristics of the questionnaires used in the study

- Are the METs = Mets values? (lines 115-117) are considered correct? where do these values come from, maybe from the publication, if so, which one?

- Why was index 3 used in BREQ?

- Did the students have 15-20 minutes to complete all three questionnaires in the same time? maybe it was one separately prepared questionnaire? (line 141-145)

- line 153 - Is the sentence correct? - The variables used were…. -  maybe better - the responses obtained from each of the questionnaires were analyzed

Results

- It is a pity that the authors did not carry out a more detailed assessment with a division into fields of study, but only a division based on gender

- I suggest that you expand the K-S abbreviation and correct this sentence accordingly (line 170) - this test does not apply to correlation.

- The digital record needs to be standardized, now in the text we have - dots and in the tables - commas.

- Is p <0.01 considered statistically significant in social science surveys?

Discussion

- Generally, the authors emphasize the convergence of the results obtained in this study with the results of previous studies. Perhaps it would be worth paying closer attention to the previously studied groups (the articles by other authors) in the context of this work - the fact is that it appears somewhere, but the uniqueness of this research group is not emphasized.

- Can it be assumed that, regardless of the age of the respondents, economic and social status, the motivations for future physical activity are similar (it is about those confirmed by statistical results)?

Conclusions

- Can the authors give examples from life - the factors with higher self-determination (line 277) and what examples can be - self-determined regulation (line 284)

 I propose to consider creating a list of abbreviations

Reviewer 4 Report

In the manuscript, the authors reported a study focusing on motivation to participate in physical exercise among college students. The findings have potential implications in promoting health behavior among young adults. However, the analyses were exploratory, and the rationale of the study needs to be further clarified. Please see below for the detailed comments:

  1. The introduction was well written and provided a comprehensive summary of the theoretical background. However, the reason why health science and medical students were chosen as the target sample was not stated. Is this sample inherently different from other populations? Because this sample only represent a small portion of the population (i.e., high SES and education level), the selection of the sample may undermine the validity of the findings.
  2. It's also needed to elaborate on why participation in PA is important for young adults, as opposed to older adults.
  3. The main issue of the paper is the lack of hypotheses, which makes the analyses exploratory. Given the literature reviewed in the introduction, several hypotheses regrading gender differences and motivation's associated with PA level could be formed.
  4. The sample was dominated by female participants. Please clarify the reason and whether gender was controlled in the analyses.
  5. METs calculated from IPAQ may also capture the non-exercise physical activity, e.g., physical stress in daily life. Thus, it's important to examine the relationship between SES and METs.
  6. The tables and figures are a bit overwhelming. Due to the lack of hypothesis-testing, there was not focus in the data presentation.
  7. Based on the results, what are the strategies that can be used to improve motivation of PA participation? This information should be specified in the Discussion.

Author Response

We appreciate all the comments and suggestions that the reviewer has made about our work, to improve its quality.

P 1: The introduction was well written and provided a comprehensive summary of the theoretical background. However, the reason why health science and medical students were chosen as the target sample was not stated. Is this sample inherently different from other populations? Because this sample only represent a small portion of the population (i.e., high SES and education level), the selection of the sample may undermine the validity of the findings.

R 1: The authors are grateful for the reviewer's appreciation. It has been decided to add in the following text, after the objective of the study, a clarification of the reason for the decision to approach the research with students in the field of Health Sciences at the present time and not to consider the comparison with the rest of the branches of knowledge. This aspect could be a very important topic to analyze in the future as a new line of research. "Focusing on this branch of knowledge allows us to know these findings after the levels of restrictions and difficulties presented by the population under study, being able to see the influence of the pandemic on the interest and performance of physical activity."

P 2: It's also needed to elaborate on why participation in PA is important for young adults, as opposed to older adults.

R 2: To explain this importance, some evidence has been included in the introduction to justify it.

P 3: The main issue of the paper is the lack of hypotheses, which makes the analyses exploratory. Given the literature reviewed in the introduction, several hypotheses regrading gender differences and motivation's associated with PA level could be formed.

R 3: Hypotheses are added to the study.

P 4: The sample was dominated by female participants. Please clarify the reason and whether gender was controlled in the analyses.

R 4: The reason why there are more women than men is because of the academic degrees that have been added to research. In this sense, in the field of Health Sciences, the rate of female students enrolled in the University, and in general, is higher than that of men, a fact that is reflected in this sample since the entire university census that was present in the classroom was used.

P 5: METs calculated from IPAQ may also capture the non-exercise physical activity, e.g., physical stress in daily life. Thus, it's important to examine the relationship between SES and METs.

R 5: All physical activity has been captured to obtain the IPAQ.

P 6: The tables and figures are a bit overwhelming. Due to the lack of hypothesis-testing, there was not focus in the data presentation.

R 6: Hypotheses have been incorporated to clarify this aspect.

P 7: Based on the results, what are the strategies that can be used to improve motivation of PA participation? This information should be specified in the Discussion.

R 7: Strategies concerning the motivation of PA participation are added.

We look forward to hearing from you.

The authors.
